# Exploring Structural Evolution of Portland Cement Blended with Supplementary Cementitious Materials in Seawater

**DOI:** 10.3390/ma14051210

**Published:** 2021-03-04

**Authors:** Solmoi Park, Jun Kil Park, Namkon Lee, Min Ook Kim

**Affiliations:** 1Department of Civil Engineering, Pukyong National University, 45 Yongso-ro, Nam-gu, Busan 48513, Korea; solmoi.park@pknu.ac.kr; 2Coastal Development and Ocean Energy Research Center, Korea Institute of Ocean Science and Technology, 385 Haeyang-ro, Yeongdo-gu, Busan 49111, Korea; jkpark@kiost.ac.kr; 3Structural Engineering Research Division, Korea Institute of Civil Engineering and Building Technology, 283 Goyangdae-ro, Ilsanseo-gu, Goyang-si, Gyeonggi-do 10223, Korea; nklee@kict.re.kr; 4Department of Civil Engineering, Seoul National University of Science and Technology, 232 Gongneung-ro, Nowon-gu, Seoul 01811, Korea

**Keywords:** Portland cement, supplementary cementitious materials, seawater, phase evolution

## Abstract

The present study investigated the structural evolution of Portland cement (PC) incorporating supplementary cementitious materials (SCMs) exposed to seawater. The samples were made with replacing Portland cement with 10 mass-% silica fume, metakaolin or glass powder. The reaction degree of SCMs estimated by the portlandite consumption shows that metakaolin has the highest reaction degree, thus metakaolin-blended PC exhibits the highest strength. The control exposed to seawater exhibited 14.82% and 12.14% higher compressive strengths compared to those cured in tap water at 7 and 28 days. The samples incorporating metakaolin showed the highest compressive strength of 76.60 MPa at 90 days tap water curing and this was 17% higher than that of the control. Exposure to seawater is found to retard the rate of hydration in all SCM-incorporating systems, while the strength development of the neat PC system is enhanced. The main reaction product that forms during exposure to seawater is Cl-AFm and brucite, while it is predicted by the thermodynamic modelling that a significant amount of M-S-H, calcite and hydrotalcite is to form at an extended period of exposure time.

## 1. Introduction

Marine and coastal concrete structures placed in a sea environment are constantly exposed to seawater, which contains various ions that interact with hydrated cement phases over time [1]. A large volume of these structures is located in the splash zone involving wetting and drying cycles, which accelerates chloride ingress into the structures. Constant exposure to seawater can be particularly deleterious for reinforced concrete, due to corrosion of embedded steel rebars. In addition, seawater also contains sulfates and carbonates, which induce precipitation of phases that are initially absent in the hydrated cement and potentially influence the ingress of chloride [2]. When concrete is exposed to seawater, it is expected to develop strength and set at a faster rate than it does at an ambient condition, due to the presence of chloride which accelerates the rate of cement hydration [3,4,5]. On the other hand, compressive strength of the concrete at later ages (e.g., after 28 days or more) can be reduced because of the high amount of sulfate that extend the process of ettringite crystallization [6,7].

Supplementary cementitious materials (SCMs)-based cement has been widely appreciated in marine concrete for its enhanced performance against chloride ingress [8,9,10]; more recently, use of SCMs as a partial replacement of Portland cement (PC) has become increasingly important, due to the CO_2_ emissions associated with production of PC involving calcination of limestone [11,12,13,14]. The permeability of concrete can be significantly improved with the addition of SCMs as they play key roles to refine micro-pores in concrete, and also modify their internal structures. Thus, the appropriate selection for both SCM type and cement replacement ratio are important to ensure the designed life-time and serviceability of marine concrete structures. The durability performance of PC incorporating metakaolin and limestone was investigated by Zhenguo et al. [15,16], which reported that metakaolin in Portland cement increases the chloride-binding capacity, attributed to the formation of more Friedel’s salt. The chloride diffusion in concrete can primarily be affected by the pore structure, hence lowering the water-to-binder ratio tends to significantly lower the chloride diffusion coefficient, while incorporation of blast furnace slag or fly ash brings the similar effect due to the enhanced chloride binding capacity [17]. In addition to the effect of SCMs on improving the resistance against chloride ingress, the hydration of slag in concrete is known to be improved when mixed with seawater [18]. Other SCMs such as glass powder are also known to positively affect the chloride resistance of concrete [19]. The effects of incorporating nanomaterials in cementitious materials to improve its material properties have been investigated [20,21,22]. Sikora et al. focused on the effect of seawater on the hydration, microstructural changes, and strength improvement in Portland cement pastes containing colloidal silica and confirmed the synergistic effect of seawater and colloidal silica [20]. Shakiba et al. investigated the microstructural change and hydration process of cement paste containing the high volume of natural pozzolan and reported the improvement of microstructure and pore filling effect [21]. Singh et al. conducted the comprehensive review work on the effect of nano-silica for the cementitious materials and pointed out many advantages of using nanomaterials for improvement of mechanical properties and durability [22]. Abousnina et al. recently studied the effect of oil-contaminated sand on the materials characteristics of cement mortar samples and confirmed the importance of appropriate mixing and curing conditions [23]. Siddika et al. reported that the better understanding of the mechanical and durability properties of cementitious materials is important for the application of 3D-printing technologies to actual structures [24].

Despite a vast number of studies having investigated the effect of SCMs on the chloride resistance of concrete, only a few have been given to observe the phase evolution of SCMs-incorporated concrete exposed to seawater. This work therefore aims to scrutinize the roles of three different SCMs including silica fume, metakaolin, and glass powder on the strength development and microstructures when exposed to actual seawater environments. To be more specific, comparisons between water- and seawater-curing conditions, and relationships between strength values measured after 1, 7, 28, and 90 days and microstructural changes were experimentally examined and compared with thermodynamic modelling results. In addition, the effects of different SCMs and seawater on the measured compressive strength were systematically investigated.

## 2. Research Significance

A great number of studies has been carried out to clarify the role of seawater for concrete casting and/or curing while the effect of seawater on the phase change of Portland cement mortar containing SCMs are not well understood. In general, it is expected that cement mortar containing SCMs might have enhanced durability and mechanical properties compared to control. However, systematic approach to determine the appropriate SCMs which can contribute the robust performance of concrete are still lacking. Thus, this study aimed to understand the phase change of cement mortars containing various SCMs and exposed to seawater to ensure improved mechanical and durability properties of concrete. To this end, an experimental program was designed, and a comprehensive experimental study was conducted to investigate the effects of three SCMs on the phase evolution of cement mortars exposed to seawater.

## 3. Materials and Methods

Mortar samples were prepared at a constant mass ratio of water:binder:sand = 0.5:1:1.1, where the binder material was either neat PC or PC blended with 10 mass-% silica fume, metakaolin or glass powder. The mix proportion was determined to have the maximum content of binders in order to allow better resolution in the characterization results. Higher sand contents made it difficult to identify phases other than quartz, which is abundantly present in the sand. Type I Portland cement (class 42.5) manufactured by Sampyo Cement (Samcheok-si, Korea) was used in this study. The sand, silica fume (micro-silica 940), metakaolin, and glass powder (MF300) were supplied by ACS corporation (Seoul, Korea), Elkem (Oslo, Norway), Nycon Materials Co., Ltd. (Asan-si, Korea), and Jiaxing Sunny FRP industries (Jiaxing, China), respectively. The samples were fabricated and cured at room temperature in a sealed condition for the initial 24 h. The samples were thereafter demolded, and were cured either at 100% relative humidity or immersed seawater. At the age of 1, 7, 28 and 90 days of curing the samples were prepared for characterization by solvent-exchanging the ground samples using isopropanol to arrest the hydration, desiccating over silica gel and further grinding to pass a 64 µm sieve. The chemical compositions obtained by X-ray fluorescence (XRF), X-ray diffraction (XRD) patterns, and the particle size distributions of the binder materials and sand are shown in Table 1 and Table 2, Figure 1 and Figure 2. Table 3 describes the chemical composition of seawater used in this study.

XRD patterns of the powdered samples were obtained using an X’Pert Pro X-ray diffractometer (Malvern Panalytical, Malvern, UK) at 30 mA and 40 kV and using CuKα radiation. The samples were scanned using an X’Celerator detector (Malvern Panalytical, Malvern, UK) at a step size of 0.026° 2θ for 2 h. The thermogravimetric analysis (TGA) was conducted using a TA Instruments Q600 instrument (PH407) (Busan, KBSI, Korea) in N_2_. A heating rate of 10 °C/min was used. The compressive strength of the samples was measured by displacement control at a rate of 1.0 mm/min as shown in Figure 3.

The thermodynamic modelling of the hydrated samples was conducted using the Gibbs energy minimization software GEM-Selektor v.3.5, coupled with CEMDATA18.01 [25]. The reaction degrees of the cement clinkers (C_3_S, C_2_S, C_3_A and C_4_AF) were obtained by correlating the amount of portlandite in the neat PC samples as quantified by TGA to that calculated by the thermodynamic modelling, which is based on the dissolution rate predicted by Parrot and Killoh’s hydration model [26]. The parameters used in the Parrot and Killoh’s hydration model were identical to those reported in [27,28]. It was assumed that the reaction degrees of clinkers are identical in other samples containing SCMs, despite that the filler effect may increase the degree of hydration of clinker minerals in SCM-incorporating samples [29]. The reaction degrees of SCMs in hydrated samples were assumed to be equivalent to the thermodynamic modelling result that predicts the amount of portlandite quantified by TGA.

## 4. Results

The reaction degrees of cement clinker minerals predicted by the thermodynamic calculation which is correlated to the actual amount formed in the neat PC samples are shown in Figure 4a. The following equation was used to fit the experimental data for cement clinkers (C_3_S, C_2_S, C_3_A and C_4_AF) and SCMs.
(1)$$\mathrm{Degree}\;\mathrm{of}\;\mathrm{reaction}\left(\%\right)=D+\left(\frac{A-D}{1+\left({\left(\frac{t}{C}\right)}^{B}\right)}\right)$$
where *A*, *B*, *C* and *D* are parameters used to simulate a curve, and *t* is time (days). The fitted parameters and the *R*^2^ values were summarized in Table 4.

It is estimated that the reaction degrees of C_3_S, C_2_S, C_3_A and C_4_AF reach 91%, 70%, 91% and 80%, respectively, after 90 days of curing. The degrees of reaction of metakaolin, silica fume and glass powder estimated using thermodynamic modelling (Figure 4b) are 96%, 61% and 65%, respectively, after 90 days of curing, showing that metakaolin is the most reactive SCM among the ones tested in this work, and the other two SCMs present a similar level of reaction kinetics. Meanwhile, portlandite consumption by glass powder predicted by the thermodynamic calculation exceeded what is expected at 100% reaction of glass powder, hence it is excluded from the analysis. This can be due to the fact that glass powder was particularly effective as a filler to induce the filler effect, enhancing the hydration of clinkers and portlandite formation.

Averaged compressive strength values of the samples are presented in Figure 5. Table 5 includes averaged compressive strength values measured at 1, 7, 28, and 90 days. It can be seen that seawater-cured samples have slower strength development except the neat PC samples. The neat PC samples exposed to seawater exhibit 14.82% and 12.14% higher compressive strengths compared to those cured in tap water cured samples after 7 and 28 days, respectively, while they show 2.12% lower strength at 90 days. The PC-metakaolin samples exhibit the highest compressive strength regardless of curing environments and curing ages, which can be attributed to the higher reaction degree of metakaolin compared to other two SCMs as mentioned previously. The samples incorporating metakaolin showed the highest compressive strength of 76.60 MPa at 90 days tap water curing and this was 17% higher than that of control. It should be noted that the effects of SCMs on compressive strength are all minimized in seawater environments, and results are well matched with experimental results of previous study [30]. Furthermore, control showed the highest variation with changing exposure condition as expected. A previous study confirmed the effectiveness of metakaolin addition in concrete to prevent strength reduction, however, this is contributed to the fact that they used NaCl solution for the curing rather than using actual seawater [31]. Figure 6 shows some representative failure modes after the measurement of compressive strength. It can be seen that Portland cement mortars (PCMs) containing silica fume (SF) or metakaolin (MK) exhibited the robust performance regardless of exposure conditions while PC and PCM containing glass powder (GP) showed more brittle failure. This can be related to the high reaction of PCMs containing SF or MK, and both samples might also have dense microstructures compared to others.

### 4.1. Phase Assemblage of Neat PC

The phase assemblage of the neat PC system during the hydration is simulated in Figure 7a. It is predicted that the hydration of clinker minerals over the hydration time mainly leads to formation of C–S–H, portlandite and ettringite, and a minor quantity of Fe-hydrogarnet and brucite. The XRD patterns of the corresponding samples in Figure 8a show the phase assemblage similar to the thermodynamic prediction, while the measured patterns also indicate the presence of hemicarbonate and monocarbonate in addition to the main hydration products predicted by the modelling. Such discrepancy between the experimental and modelling result can be due to that the stability of ettringite is overestimated in the modelling or can also be associated with the dissolution of calcite predicted by the modelling. The derivative thermogravimetric analysis (DTG) curves of the neat PC samples during hydration shown in Figure 9a are in close agreement with the XRD results, showing that hydration products C–S–H, portlandite, AFt and AFm phases increasingly form over the hydration time.

Exposure to seawater is expected to initially deplete portlandite in the neat PC system, followed by the destabilization of ettringite and C–S–H, and precipitation of brucite (Figure 7b). Gypsum is predicted to form as a transient phase during exposure to seawater, while the final products predicted to stabilize are M-S-H, hydrotalcite, calcite and brucite. It is seen that the modelling predicts no AFm phase formation during exposure to seawater, while the XRD patterns of the samples show that Cl-AFm dominantly forms when immersed in seawater (Figure 8b). This tendency is also reflected in the DTG curves of the samples immersed in seawater shown in Figure 9b, which suggests that AFm phases increasingly form over time during the immersion in seawater.

### 4.2. Phase Assemblage of PC-Silica Fume

The predicted hydration phase assemblage of the PC-silica fume system shown in Figure 7c suggests that silica fume incorporation leads to consumption of portlandite, hence the volume of portlandite is lower in this system compared to that in the neat PC, and formation of more C–S–H. The volume of portlandite is predicted to increase up to the age of 1 day, and it then decreases as the dissolution of silica fume occurs.

The thermodynamic calculation predicts no AFm-related phase would form, despite that the XRD patterns in Figure 8c shows that hemicarbonate and monocarbonate form. The intensity of the peak associated with the presence of hemicarbonate is particularly notable and comparable to that of monocarbonate. This observation implies that the AFm phase that dominantly forms in the neat PC is monocarbonate, while silica fume incorporation allows hemicarbonate to notably form. Meanwhile, the DTG curves of both systems (Figure 9a,c) indicate that AFm phases increasingly form after 1 day.

The destabilization of hydration products (C–S–H, portlandite and ettringite) in the PC-silica fume upon exposure to seawater is similar to that in the neat PC system, while it is predicted that less brucite and more M-S-H form in the PC-silica fume system (Figure 7d). The XRD result (Figure 8d) and the DTG curves (Figure 9d) show that brucite and hydrotalcite are stable during immersion in seawater for 90 days, despite that the modelling result predicts the stability of hydrotalcite is met once all ettringite is depleted. In addition, the it is observed that the mass loss due to C–S–H, ettringite and AFm-related phases continuously increase during immersion in seawater according to the TG results in Figure 9d, indicating that the hydration of PC clinkers ceaselessly occur in seawater. The modelling result in this work could not be based on the actual quantity of reacted clinkers and uptake of seawater, hence, it should only be understood as what is generally expected as a function of seawater uptake by the modelled system.

### 4.3. Phase Assemblage of PC-Metakaolin

The hydration phase assemblage of the PC-metakaolin system is predicted in Figure 7e. The modelling result shows that brucite is initially predicted stable until it is destabilized to hydrotalcite due to the dissolution of metakaolin occurs at the age of ~1 day. It is also predicted that ettringite significantly forms until this age, then Fe-hydrogarnet and monocarbonate form as a secondary product. The XRD result in Figure 8e shows that monocarbonate forms first then hemicarbonate is stabilized in the PC-metakaolin system as similar to the PC-silica fume one. The peak in the DTG curve corresponding to the dehydration of AFm-related phases in Figure 9e is more notable than it is in the previous two systems after 7 days of curing.

Immersion of the PC-metakaolin system in seawater is predicted to lead to destabilization of monocarbonate to Friedel’s salt, which later destabilizes to ettringite (Figure 7f). The amount of ettringite formed in this system is expected to increase during immersion in seawater, along with brucite which is temporarily stable and destabilizes to hydrotalcite. In addition, the modelling result for the system immersed in seawater predicts gypsum to form as a transient phase, which destabilizes to calcite. The XRD and TGA results in Figure 8f or Figure 9f show the hydration phase assemblages in the hydrated and immersed samples are similar, except for hydrocalumite observed in the XRD result for the sample immersed in seawater. This is probably because the immersion time taken for the sample has not been sufficient to exhibit the phase transition simulated by the thermodynamic modelling. In addition, the presence of brucite is observed in the immersed sample as early as 7 days of the sample age (Figure 9f), hence, brucite is a more stable phase in the immersed condition than it is predicted by the modelling.

### 4.4. Phase Assemblage of PC-Glass Powder

The phase assemblage of the PC-glass powder system during 90 days of hydration is predicted in Figure 7g. Although the glass powder used in this work shows a reaction degree similar to that of the silica fume, it consumes much less portlandite. The modelling results predict the formation of C–S–H, ettringite and portlandite as major products and Fe-hydrogarnet, brucite and hydrotalcite as minor products. It is observed in XRD patterns for the hydrated samples shown in Figure 8g that monocarbonate is dominantly present among other AFm phases after 7 and 28 days, and hemicarbonate forms at later ages. Immersion in seawater is expected to lead to destabilization of hydrotalcite, which is initially present in this system, to brucite. Upon depletion of ettringite there forms gypsum as a transient phase. Calcite is expected to notably precipitate when gypsum is no longer a stable phase, as similar to other systems that were simulated by the thermodynamic modelling. The mass loss due to the dehydration of AFm phases and hydrotalcite in the DTG curve (Figure 9h) is significant after 28 days of immersion, indicating that these two phases notably form during the 7–28 days age.

## 5. Discussion

In this study, the phase changes in Portland cement mortars containing three different SCMs such as silica fume, metakaolin, and glass powder were experimentally investigated using XRD, TGA analysis and, thermodynamic modelling was conducted to see the reaction degrees of each SCM. The effect of seawater exposure on both phase changes and strength development of samples was systematically investigated and compared. It was originally expected that the mortar sample with silica fume will show the highest reaction degree and faster strength development due to its high pozzolanic reaction and large surface area. However, sample with metakaolin exhibited the highest reaction degree compared to other SCMs and this was also confirmed from the measured compressive strength values. As expected, mortar sample with glass powder exhibited the slow strength development and this might be related to slow pozzolanic reaction as pointed out previously. Based on the obtained results and comparisons, it can be concluded that metakaolin is promising supplementary cementitious material for the marine and coastal concrete structures to ensure early strength gain. The results from this study might be beneficial for researchers and/or engineers who working for marine and coastal concrete structures. Additionally, this is because SCM should be incorporated in marine concrete to ensure better durability and to ensure designed service life of marine and coastal structures. However, it should be noted that this study focused on the phase assemblage in mortar samples during at the early stages between 1 and 90 days, thus, further research might be necessary to investigate the long-term behavior of cementitious composites exposed to seawater environment. Furthermore, durability and mechanical properties (e.g., tensile strength) of mortar samples exposed to such condition must be investigated to provide better information for the scientists and engineers who working on the maintenance and construction of marine and coastal concrete structures.

## 6. Conclusions

The present study investigated the structural evolution of SCM-incorporating cement in seawater. Based on the test results and comparisons, the following conclusions can be drawn:The strength development of the neat PC samples was faster when exposed to seawater, while other samples incorporating SCMs exhibited the slower strength development.The control exposed to seawater exhibited 14.82% and 12.14% higher compressive strengths compared to those cured in tap water at 7 and 28 days.The samples incorporating metakaolin showed the highest compressive strength of 76.60 MPa at 90 days tap water curing and this was 17% higher than that of control.The obtained characterization and modelling results show that the phase assemblages of the samples are similar, while there is a quantitative difference in the AFm-related phases.The amount of portlandite is highest in the PC-glass powder system due to the high Ca content of the glass powder.The main reaction product experimentally observed during the immersion in seawater over 90 days is Cl-AFm and brucite.The modelling results predict that M-S-H, calcite and hydrotalcite are to increasingly form at an extended timescale.

## Figures and Tables

**Figure 1 materials-14-01210-f001:**
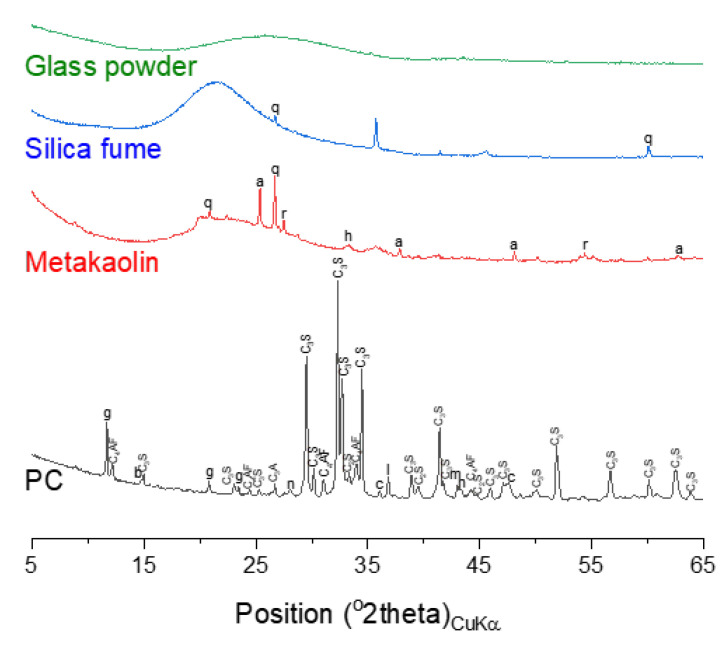
X-ray diffraction (XRD) patterns of raw materials. The following symbols are used to denote: a—anatase, b—bassanite, c—calcite, g—gypsum, h—hematite, n—thernadite, m—periclase, q—quartz, r—rutile.

**Figure 2 materials-14-01210-f002:**
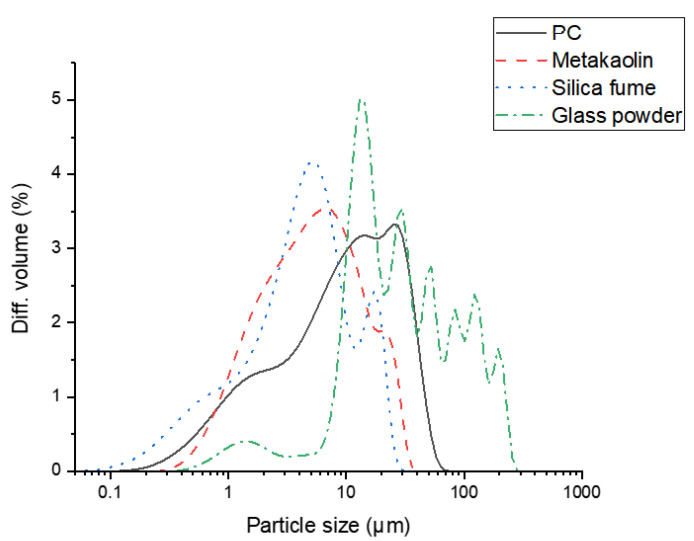
Particle size distributions of raw materials obtained by laser diffraction.

**Figure 3 materials-14-01210-f003:**
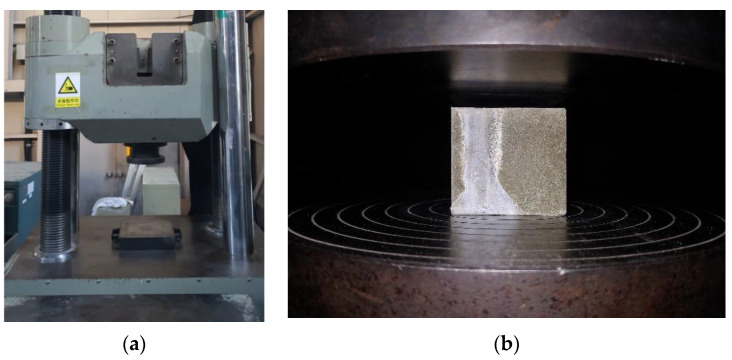
Compressive strength measurement: (**a**) Universal testing machine; (**b**) Compressive strength test setup.

**Figure 4 materials-14-01210-f004:**
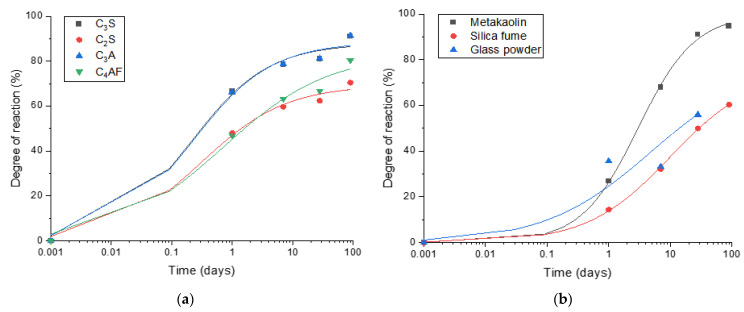
The reaction degrees of cement clinkers and SCMs using the amount of portlandite obtained by TGA and thermodynamic calculations: (**a**) Reaction degrees of cement clinkers predicted by thermodynamic calculation; (**b**) Reaction degrees of SCMs estimated using thermodynamic modelling.

**Figure 5 materials-14-01210-f005:**
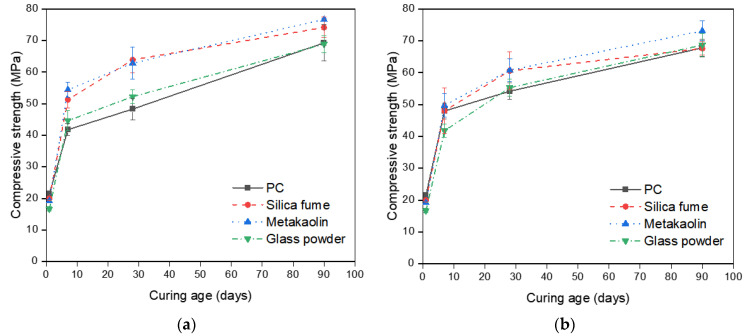
Compressive strength development of samples: (**a**) Measured strength values of samples cured in tap water; (**b**) Measured strength values of samples cured in seawater.

**Figure 6 materials-14-01210-f006:**
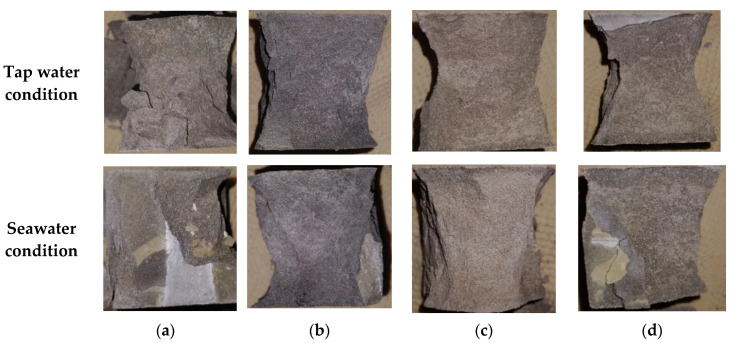
Representative failure modes of samples after 28 days compressive strength measurement: (**a**) neat PC; (**b**) PCM containing silica fume; (**c**) PCM containing metakaolin; (**d**) PCM containing class powder.

**Figure 7 materials-14-01210-f007:**
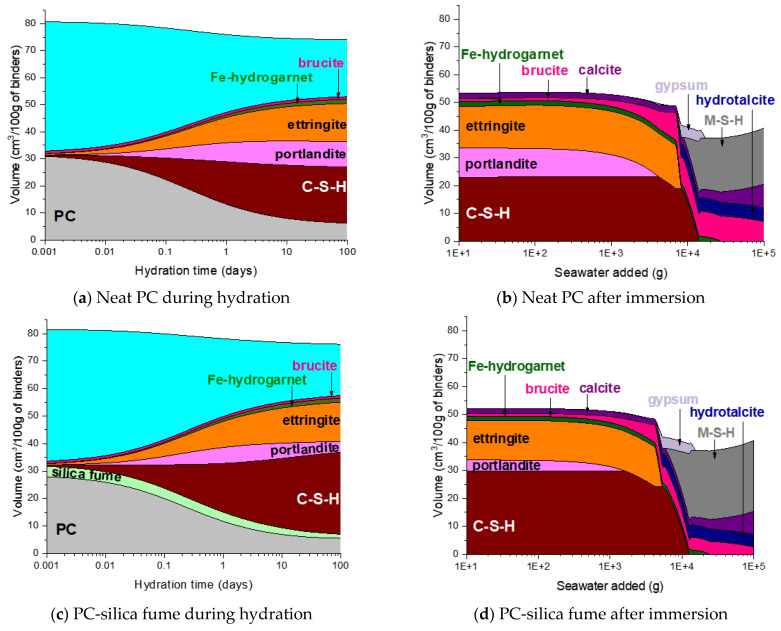
Predicted phase assemblages of neat PC and PC blended with silica fume, metakaolin or glass powder during hydration and after immersion in seawater.

**Figure 8 materials-14-01210-f008:**
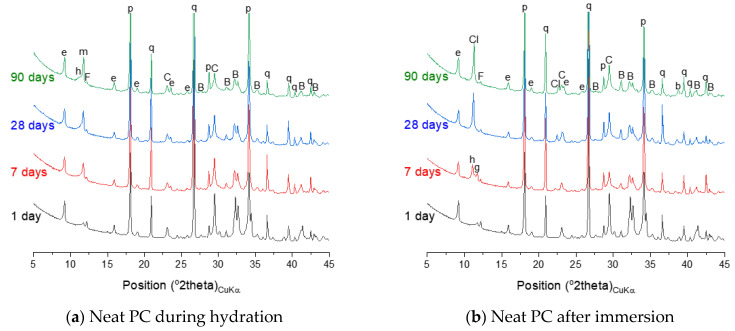
X-ray diffraction (XRD) patterns of neat PC and PC blended with silica fume, metakaolin or glass powder during hydration and after immersion in seawater up to 90 days. The following symbols are used to denote: B—belite, C—calcite, F—ferrite, q—quartz, b—brucite, e—ettringite, h—hemicarbonate, m—monocarbonate, p—portlandite, Cl—hydrocalumite.

**Figure 9 materials-14-01210-f009:**
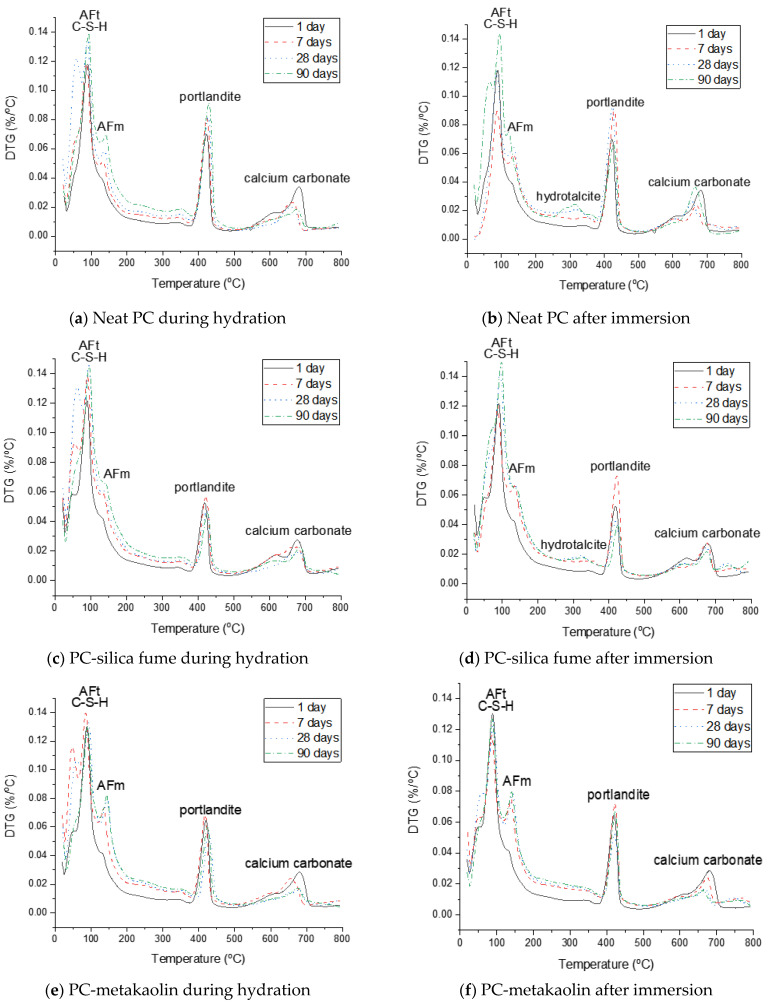
Derivative thermogravimetric analysis (DTG) curves of neat PC and PC blended with silica fume, metakaolin or glass powder during hydration and after immersion in seawater up to 90 days.

**Table 1 materials-14-01210-t001:** Oxide compositions of raw materials obtained by X-ray fluorescence analysis.

Composition	Cement	Silica Fume	Metakaolin	Glass Powder
CaO	61.0	0.2	0.6	30.1
SiO_2_	17.2	92.8	48.5	51.9
Al_2_O_3_	4.8	0.2	43.4	13.5
Fe_2_O_3_	4.1	2.2	3.2	0.3
SO_3_	3.3	0.6	0.1	-
Na_2_O	0.2	0.2	0.1	0.6
K_2_O	1.2	0.3	0.3	0.1
MgO	2.4	0.2	0.1	1.2
SrO	0.1	-	-	0.4
TiO_2_	0.4	-	2.3	0.4
Others	0.9	0.2	0.4	0.5
LOI	4.4	3.1	1.1	1.1

**Table 2 materials-14-01210-t002:** Sieve analysis of sand used.

Sieve Size	Mass Retained (%)	Cumulative Mass Retained (%)
1.0 mm	0.00	0.00
850 µm	0.01	0.01
600 µm	1.20	1.21
425 µm	6.50	7.71
300 µm	28.00	35.71
212 µm	33.94	69.65
150 µm	21.05	90.70
106 µm	7.50	98.20
75 µm	1.50	99.70
Pan	0.30	100.00

**Table 3 materials-14-01210-t003:** Chemical composition of seawater.

Ions	Concentration (mg/L)
Cl^−^	21,075 ± 829
Br^−^	51 ± 2
SO_4_^2−^	2258 ± 147
Na^+^	17,075 ± 1798
K^+^	549 ± 40
Ca^2+^	364 ±12
Mg^2+^	973 ± 41

**Table 4 materials-14-01210-t004:** The fitted parameters for simulating the curves shown in Figure 3.

Parameters	*A*	*B*	*C*	*D*	*R* ^2^
C3S	0	0.676	0.205	87.878	0.991
C2S	0	0.616	0.300	69.407	0.992
C3A	0	0.665	0.218	88.502	0.991
C4AF	0	0.498	0.707	83.434	0.986
Metakaolin	0.082	0.941	2.975	99.844	0.999
Silica fume	0	0.644	10	75.111	0.999
Glass powder *	0.008	0.5	5	80	-

* The fit did not converge for glass powder.

**Table 5 materials-14-01210-t005:** Measured compressive strength.

Type	Compressive Strength (MPa)							
	Tap Water				Seawater			
	1 Day	7 Days	28 Days	90 Days	1 Day	7 Days	28 Days	90 Days
Portland Cement	21.27	41.81	50.88	65.33	21.27	46.53	52.09	58.70
Silica Fume	20.48	53.12	64.42	71.25	20.48	54.61	62.13	67.58
Metakaolin	19.51	52.84	66.77	76.60	19.51	52.83	61.27	74.75
Glass Powder	16.49	46.94	53.79	66.67	16.49	41.02	53.12	65.81

## Data Availability

Data sharing is not applicable to this article.

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
