# Peer review of "Exploring Structural Evolution of Portland Cement Blended with Supplementary Cementitious Materials in Seawater"

_materials, 2021, doi:10.3390/ma14051210_

Round 1
Reviewer 1 Report
The manuscript titled Exploring Structural Evolution of Portland Cement Blended with Supplementary Cementitious Materials in Seawater presents an experimental study investigating the structural modifications of Portland Cement with various supplementary materials when exposed to seawater.
The experiments were carefully conducted and the data was processed and interpreted. However, there is one observation the authors should address before the manuscript can be published.
Figure 3 a and b shows the degree of reaction versus time; it is not clear how these plots were obtained. It appears that a straight segment exists from 0.001 to approximately 0.1 and then a discontinuity occurs. The authors should provide more details on what happens there. The authors should mention what type of correlation was used and what were the parameter of the regression model.
Reviewer 2 Report
The article covers the topic of the Exploring Structural Evolution of Portland Cement Blended with Supplementary Cementitious Materials in Seawater. In my opinion, the authors have provided interesting research work.
The manuscript has acceptable cohesion.
The subject is informative and present added value to the body of knowledge on the subject area.
However, some modification should be considered:
1. The abstract should contain major results obtained in the research (containing received values).
2. In my opinion, introduction part should be extended.
3. I suggest to add separated point 2 - Research significance - Please describe here the main essence of the research. (Why presented
paper is so important? What is major innovation accent in presented studies?). A part of text in lines 59-69 could be useful.
4. The content of the mixes is surprising. The authors write that: 'Mortar samples were prepared at a constant mass ratio of
water:binder:sand = 0.5:1:1.1.' Why this content was chosen? In general, to examine cement mortar the following mass content is using: water:binder:sand = 0.5:1:3.
Please clearly determine why do you propose mentioned cement mortar content and convince me that your choice was necessary.
5. What was the source of the cement, sand, and other components? Where do these products come from? Please provide details ex. manufacturer, localization.
6. Please add the grain size curve of components (especially for sand and cement).
7. What type of cement was used in this research? Please provide cement class (32,5 or 42,5 or 52,5 or another?).
Please add information, if cement contain any addition, or it is clear portland cement.
8. Please ensure description and photo of testing machine (evaluation of compressive strength) which was used in this surveys.
What is more, please show all tested specimens (photos).
9. It is strongly recommended to indicate potential application of research results in civil engineering.
10. Conclusion needs more elaboration. Please paraphrase your results and discussions and use them in the conclusion part.
Please use more sentences containing percentages and illustrate the main conclusions in the manuscript.
Reviewer 3 Report
The review of the paper entitled “Exploring Structural Evolution of Portland Cement Blended with Supplementary Cementitious Materials in Seawater” is done. In this research the authors studied the structural evolution of Portland cement with supplementary cementitious materials. This is an interesting study with both experimental and computational approaches and the authors have tried to explain the results and the methodology with details. It can be published in the Materials Journal with the following comments:
- Nowadays nanomaterials are very popular in improving the properties of different types of concrete such as strength and also their durability against different ions in seawater for example nano-silica and nano-TiO2 just to name a few. There is an active area of research in this field of science for concrete technology. Therefore, it definitely worth to add a brief paragraph about this topic in the Introduction where it doesn’t slow the flow of the paper. You can use the following references or any other references that could be useful for this topic. The recommended references are:
Sikora, P., Cendrowski, K., Abd Elrahman, M., Chung, S.Y., Mijowska, E. and Stephan, D., 2019. The effects of seawater on the hydration, microstructure and strength development of Portland cement pastes incorporating colloidal silica. Applied Nanoscience, pp.1-12.
Shakiba, M., Rahgozar, P., Elahi, A.R. and Rahgozar, R., 2018. Effect of activated pozzolan with Ca(OH)2 and nano-SiO2 on microstructure and hydration of high-volume natural pozzolan paste. Civil Engineering Journal, 4(10), pp.2437-2449.
Singh, L.P., Karade, S.R., Bhattacharyya, S.K., Yousuf, M.M. and Ahalawat, S., 2013. Beneficial role of nanosilica in cement based materials–A review. Construction and Building Materials, 47, pp.1069-1077.
- Numerous number of Figures slow the flow of the paper and the comparison between the used materials in this study is harder to be observed. These Figures should be merged by juxtaposing them. Please merge Figures 5, 8, 11, and 14 for the predicted phase assemblages, Figures 6, 9, 12, and 15 for the XRD patterns, and Figures 7, 10, 13, and 16 for DTG curves. Also, please fix the Figure numbers in the text accordingly.
- Please extend the Discussion section and bring some parts of the Results into this section or if possible merge these two sections. If the latter is not possible due to the journal template, just note that the Results section should only contain the results and what is observed with no extra discussion. The discussion on the results should be done in the Discussion section.
- For the Author Contribution part, please put the name of each author and write the corresponding contribution in front of his/her name.
Reviewer 4 Report
Comments
This paper investigated the structural evolution of Portland cement incorporating supplementary cementitious materials exposed to seawater. The outcome is interesting for readers. However, there are several aspects that need to be improved. The reviewer can only recommend for publication if the author satisfactorily address the following comments in the revised version.
- The mechanism behind the variation of strength for PC, silica fume, metakaolin and glass powder need to be explained.
- The novelty of the study should be highlighted in the end of introduction section. How this study is different from the published study in literature?
- How the outcome of this study will benefit researchers and end users? This need to be highlighted in introduction or end of conclusion.
- The background study on the strength development process of industrial by-product concrete is insufficient. The growing interest of waste materials or industrial by-product in concrete construction need to be highlighted. The recent investigation on the strength development process in normal concrete [Ref: Characteristics, strength development and microstructure of cement mortar containing oil-contaminated sand] and 3D-printed concrete [Ref: 3D-printed concrete: applications, performance, and challenges] are particularly important. Suggest to include them in introduction section with proper citations to improve the background study.
I would be happy to see the revised version to understand how these comments are being addressed.
Round 2
Reviewer 2 Report
All remarks have been considered by authors. Errors have been eliminated. The authors responded to all comments of the reviewer.
The current version is satisfactory for reviewer.
Reviewer 4 Report
I have no further comments